# Antioxidant, Antibacterial Properties of Novel Peptide CP by Enzymatic Hydrolysis of *Chromis notata* By-Products and Its Efficacy on Atopic Dermatitis

**DOI:** 10.3390/md22010044

**Published:** 2024-01-12

**Authors:** Jin-Woo Hwang, Sung-Gyu Lee, Hyun Kang

**Affiliations:** Department of Medical Laboratory Science, College of Health Science, Dankook University, Cheonan-si 31116, Chungcheongnam-do, Republic of Korea; croucard@naver.com

**Keywords:** *Chromis notata*, by-product hydrolysate, antioxidant peptide, antimicrobial activity, atopic dermatitis

## Abstract

This study investigated the antioxidant, antimicrobial, and anti-atopic dermatitis (AD) effects of a novel peptide (CP) derived from a *Chromis notata* by-product hydrolysate. Alcalase, Flavourzyme, Neutrase, and Protamex enzymes were used to hydrolyze the *C. notata* by-product protein, and the 2,2′-azino-bis(3-ethylbenzothiazoline-6-sulfonic acid) (ABTS) radical-scavenging activity was measured. Alcalase hydrolysate exhibited the highest ABTS radical-scavenging activity, leading to the selection of Alcalase for further purification. The CHAO-1-I fraction, with the highest ABTS activity, was isolated and further purified, resulting in the identification of the peptide CP with the amino acid sequence Ala-Gln-Val-Met-Lys-Leu-Pro-His-Arg-Met-Gln-His-Ser-Gln-Ser. CP demonstrated antimicrobial activity against *Staphylococcus aureus*, inhibiting its growth. In a 2,4-dinitrochlorobenzene (DNCB)-induced AD-like skin model in mice, CP significantly alleviated skin lesions, reduced epidermal and dermal thickness, and inhibited mast cell infiltration. Moreover, CP suppressed the elevated levels of interleukin-6 (IL-6) in the plasma of DNCB-induced mice. These findings highlight the potential of CP as a therapeutic agent for AD and suggest a novel application of this *C. notata* by-product in the fish processing industry.

## 1. Introduction

Atopic dermatitis (AD) is a common, chronic, and inflammatory skin disease characterized by itching and recurrent eczematous lesions [1]. The complex interplay of susceptibility genes encoding skin barrier molecules, inflammatory response elements, environmental factors, and infectious agents, particularly *Staphylococcus aureus*, leads to an imbalance in the microbial community residing on AD skin [2,3]. This, along with the altered immune state of the host, plays a crucial role in the pathophysiology of AD [4,5,6]. The majority of patients with AD exhibit clusters of high-density *S. aureus* on both affected and unaffected skin areas. The presence of *S. aureus* is known to be associated with the exacerbation of AD, and concerns have arisen due to the prevalence of methicillin-resistant strains (MRSA), indicating antibiotic resistance [7,8].

Although the etiology of AD is not fully understood, recent advancements have shed light on oxidative stress as a contributing factor. Oxidative stress, defined by an imbalance between the generation and removal of reactive oxygen species (ROS), is implicated in various diseases, including psoriasis, asthma, cystic fibrosis, neurodegenerative disorders, and cancer [9,10,11,12,13]. Evidence suggests that oxidative stress plays a role in AD, and it is positively correlated with disease activity [14,15,16,17].

With the development of the fishing and aquaculture industries, there has been a substantial increase in the production of seafood, leading to the generation of large amounts of by-products such as fish skin, bones, and scales [18,19]. Approximately 55% to 65% of the total weight of fish catches is estimated to be discarded as by-products, posing serious environmental issues when disposed of on land or in the oceans [20,21]. Recent research has focused on exploring high-value-added materials through the discovery of functional ingredients and bioactive compounds from fishery by-products [22,23,24,25,26,27,28,29].

Peptides generated by protein digestion exhibit biological activity, and their release occurs during gastrointestinal digestion or food processing [30]. Once liberated, these bioactive peptides can manifest various physiological activities based on their structure, composition, and sequence [31]. Consequently, there is a growing interest in utilizing these peptides as functional food and pharmaceutical ingredients for maintaining health [32]. The functional properties of proteins can be enhanced through controlled enzymatic hydrolysis under specific conditions [33]. Several studies have reported the potential use of antioxidant peptides extracted from marine organisms and their application as alternative antioxidants [23,34,35,36].

*Chromis notata*, a tropical fish species distributed along the coasts of South Korea, Jeju Island, Japan, and the East China Sea, is known for its representation in tropical and subtropical regions [37]. While foundational research on *C. notata* has primarily focused on morphological characteristics [37], molecular phylogenetics [38], spawning times, and reproductive seasons [39], functional studies using *C. notata* remain scarce.

The quantity of by-products from fishery processing varies depending on the species, size, season, and fishing grounds [40,41,42]. Approximately 41.5% of fish constitute by-products, including heads, frames, internal organs, opercula, trimmings, blood, and skin, with only 15% of these by-products considered suitable for human consumption [43]. In the case of *C. notata*, parts such as the head, tail, bones, etc., excluding the trunk, remain underutilized. Leveraging these by-products of *C. notata* can minimize waste and pollution, thereby reducing environmental impacts and enhancing economic benefits in the industry [44,45]. Notably, *C. notata* heads contain bioactive proteins and peptides [43]. Interestingly, many researchers emphasize the recovery of bioactive peptides from fish excrement/by-products due to their high-quality proteins, proving the potential of these by-products as a candidate source for harvesting bioactive peptides [29].

Therefore, this study aimed to isolate novel antioxidant peptides from the head portion of *C. notata* and verify their antioxidant and anti-atopic efficacy in a 2,4-dinitrochlorobenzene (DNCB)-induced AD animal model.

## 2. Results

### 2.1. Yield of Hydrolysates from Chromis notata By-Product Protein and Selection Based on ABTS Radical Scavenging

The results of the yield of hydrolysates and 2,2′-azino-bis(3-ethylbenzothiazoline-6-sulfonic acid) (ABTS) radical-scavenging activity from *C. notata* head protein are presented in Table 1. Using Alcalase, Flavourzyme, Neutrase, and Protamex enzymes, the yield of hydrolysates from *C. notata* head protein was 68.72%, 34.14%, 49.71%, and 61.31%, respectively, with Alcalase showing the highest yield. The ABTS radical-scavenging activity was measured to evaluate antioxidant activity. The RC_50_ values, representing a 50% reduction in the concentration of radicals, were 102.62 ± 5.63, 260.48 ± 3.40, 605.61 ± 220.58, and 175.64 ± 6.98 µg/mL for Alcalase, Flavourzyme, Neutrase, and Protamex hydrolysates, respectively. Among them, the Alcalase hydrolysate exhibited the most significant ABTS radical-scavenging activity and was selected for further investigation.

### 2.2. Purification of Antioxidant Peptides from C. notata By-Product Alcalase Hydrolysate

The peptide separation process was carried out using the Alcalase hydrolysate from the head of *C. notata*, which exhibited the highest ABTS radical-scavenging activity. In the first step, separation was performed using a 3500 Da dialysis membrane to divide the sample into the inner (CHAI: *C. notata* Head Alcalase Inner membrane, ≥3500 Da) and outer (CHAO: *C. notata* Head Alcalase Outer membrane, <3500 Da) membranes. The ABTS radical RC_50_ values were determined as 207.61 and 82.14 μg/mL for the CHAI and CHAO fractions, respectively. CHAO, showing the highest ABTS radical-inhibition activity, was selected for further investigation (Figure 1A). CHAO underwent a separation process using fast protein liquid chromatography (FPLC) equipped with a GPC column. Three fractions, CHAO-1, CHAO-2, and CHAO-3, were collected, and their ABTS radical-scavenging abilities were measured. Among them, CHAO-1 demonstrated the highest activity with an RC_50_ value of 32.14 μg/mL, making it the chosen fraction for the next step (Figure 1B). Further separation of CHAO-1 was performed using high-performance liquid chromatography (HPLC) with a C_18_ column, resulting in two sub-fractions, CHAO-1-I and CHAO-1-II. Measurement of the ABTS inhibition activity for these sub-fractions revealed that CHAO-1-I had the highest antioxidant activity with an RC_50_ value of 5.12 μg/mL (Figure 1C). To obtain pure peptides, CHAO-1-I underwent purification using HPLC equipped with a GPC column (Figure 1D). The amino acid sequence of the purified peptide, as determined by the Milligen 6600 protein sequencer, was Ala-Gln-Val-Met-Lys-Leu-Pro-His-Arg-Met-Gln-His-Ser-Gln-Ser. This peptide was named CP (*C. notata* By-Product Peptide). The confirmed peptide was synthesized by A&Pep Co., Ltd. (Cheongju-si, Chungcheongbuk-do, Republic of Korea) for antimicrobial and animal experiments.

### 2.3. Structural Characterization of C. notata By-Product-Derived CP Peptide

Figure 2A shows the 3D virtual structure of the CP peptide. Figure 2B shows the synthesis results of the CP peptide, the purity of which was over 98%. The wheel and net projections have been proposed to represent in two dimensions the tridimensional helical structures and facilitate the observation of their properties, especially in terms of residue polarity and intramolecular bonding (Figure 2C). Net projections are used for the same structures as wheels but provide a different perspective to the visualization of the helixes (Figure 2C, left panel). At least two chains must be specified as a homodimer (Figure 2D). These results suggest that CP peptides are likely to adopt α-helix-dominated conformations upon interactions with bacterial biomembranes, and the antimicrobial action of the peptides could be anticipated via their amphipathic helical properties.

### 2.4. Antimicrobial Activity of CP

To assess the antimicrobial activity of CP, an agar disk diffusion assay was conducted (Figure 3A). The diameter values of the inhibition zones indicated that CP could inhibit the growth of *S. aureus*. The diameter of the inhibition zones at CP concentrations of 1, 2, and 4 mM was approximately 9, 11, and 16 mm, respectively. The broad-spectrum antibiotic penicillin was used as a positive control. The growth of *S. aureus* at various CP concentrations (1.25, 2.5, and 5 mM) was observed, and the bacterial density measured at 490 nm was found to correlate with the antimicrobial effect of CP (Figure 3B). As shown in Figure 2B, in the absence of CP (Control), *S. aureus* exhibited a normal growth curve representing lag, exponential, stationary, and decline growth phases within 24 h. With the addition of CP at 1.25 and 2.5 mM, the late exponential growth phase of the *S. aureus* growth curve was delayed. Upon treatment with 5 mM CP, a significant inhibition of *S. aureus* growth was observed within 24 h.

### 2.5. Body Weight and Histopathological Effects of CP in DNCB-Induced AD Mouse Model

A DNCB-induced AD mouse model was established through a two-step process. In the first step, skin sensitization was induced by applying 1% DNCB to mouse skin. Subsequently, in the second step, repeated applications of 0.5% DNCB were performed. On the final day, which was the 29th day, there were no significant differences in body weight among the groups, as illustrated in Figure 4A. To evaluate the effects of CP in alleviating skin lesions observed in AD, a DNCB-induced AD mouse model was employed. The reduction in erythema, inflammation, and hyperkeratosis by CP treatment for 21 days was distinctly observed in the CP-treated group compared to the DNCB group (Figure 4B). Mouse skin was stained with Masson’s Trichrome to assess the impact of CP on skin and epidermal thickness (Figure 4C). The thickness of the epidermis and dermis was measured using ImageJ software (Version 1.54h). The thickness of the epidermis and dermis significantly increased in the DNCB group compared to the normal group (## *p* < 0.01). In contrast, the thickness of the epidermis and dermis significantly decreased in the CP group compared to the DNCB group (** *p* < 0.01) (Figure 4E,F). To measure the extent of mast cell infiltration, skin tissues were stained with toluidine blue (Figure 4D). Mast cell infiltration significantly increased in the DNCB group compared to the normal group (## *p* < 0.01). The number of mast cells significantly decreased in the CP group compared to the DNCB group (** *p* < 0.01) (Figure 4G). Notably, mast cell infiltration showed a superior effect in the CP group compared to the positive control Terfenadine group (* *p* < 0.05).

### 2.6. Effects of CP on Serum IL-6 Levels in the DNCB-Induced AD Mouse Model

The impact of CP on interleukin-6 (IL-6) levels in mouse serum was assessed using an enzyme-linked immunosorbent assay (ELISA). Results revealed that DNCB treatment significantly increased (## *p* < 0.01) IL-6 levels compared to the normal group. However, CP exhibited a substantial inhibitory effect on the elevated IL-6 levels induced by DNCB (Figure 5). These findings suggest that CP attenuated mast cell activation, reduced skin thickness in AD-like mouse skin, and lowered cytokine levels in the blood induced by DNCB.

## 3. Discussion

The amount of seafood by-products discarded in the seafood processing industry ranges from 50% to 75% of the total weight, varying with species, size, aquaculture conditions, and seasons [39]. Utilizing fish by-products presents significant opportunities for the seafood processing industry, providing a chance to develop substantial production opportunities. This not only has the potential to generate additional income but also reduces disposal costs and prevents pollution [46]. Fish by-products contain proteins, fats, and amino acids, which can be obtained through protease hydrolysis [47]. Hydrolysates of fish proteins have proven to be a major source of bioactive peptides, showing promise in nutritional and pharmaceutical applications. Importantly, this approach is not only cost-effective but has also been demonstrated to minimize pollution resulting from fish waste [29,48].

In this study, we investigated the antioxidative, antimicrobial, and anti-AD effects of a novel peptide, CP, derived from *C. notata* by-product head hydrolysate. Initially, Alcalase hydrolysate, which exhibited the highest antioxidant effect among the four hydrolysates tested, was chosen for peptide isolation. Alcalase, used in the production of Alcalase hydrolysate, primarily cleaves proteins in the middle of the amino acid chain [8,9]. Its broad range of recognizable amino acids catalyzes protein hydrolysis reactions, producing hydrolysates with many small peptides known for their higher antioxidative potential and other physiological activities across various tissues [49]. Consistent with previous reports, our results demonstrated superior antioxidative activity in the CHAO < 3500 Da fraction (Figure 1A).

Subsequently, through separation and purification processes, we selected the CHAO-1-I fraction, exhibiting the highest antioxidative activity, and obtained the pure peptide CP with the amino acid sequence Ala-Gln-Val-Met-Lys-Leu-Pro-His-Arg-Met-Gln-His-Ser-Gln-Ser. Biologically active peptides are defined by specific protein fragments that can positively impact bodily functions or conditions, ultimately influencing health [50]. Short peptides, often characterized by their multifunctional properties, are reported to significantly affect absorption across the intestinal epithelium and utilization in targeted tissues [51]. Our findings support previous reports, showing higher antioxidative activity in the CHAO < 3500 Da fraction (Figure 1A). The size of the active sequence influences both the antioxidative potential and other physiological properties of peptides. The size of active sequences can vary from two to twenty amino acid residues, with many peptides exhibiting multifunctional characteristics [52]. Antioxidative activity has been associated not only with specific amino acid sequences but also with the presence of highly hydrophobic amino acids such as Leu, Met, and Ile [53]. Particularly, Met has been recognized for its exceptional antioxidative effects [54]. Therefore, we attribute the antioxidative activity of CP to the inclusion of Met, which contributes to its high activity.

Helical wheels are a standard way to predict protein sequence segments with either helical or non-helical potential. Helices are one of the most common secondary structures found in peptides and proteins. The wheel projection is mostly useful for visualizing the helix regions by peptide properties, namely acidity/basicity and the ability to form hydrophilic (hydrogen) or hydrophobic bonds [55]. To graphically represent these intramolecular interactions, different projections of the secondary structures of peptides have been created, most notably the wheel (Schiffer–Edmundson) and net projections [56].

Antimicrobial peptides (AMPs) are emerging as promising novel antimicrobials due to their mechanisms involving interactions with bacterial cell walls and membranes [57]. AMPs can be generated through various methods, including chemical modification [58], microbial fermentation [59], and enzymatic protein hydrolysis [60]. Alcalase has been utilized to produce biologically active peptides from diverse sources, such as bovine skeletal muscle proteins [61], whey proteins [62], and canola proteins [63], known to release smaller active subunits with enhanced antimicrobial activity by cutting high-molecular-weight proteins [64,65]. This optimization generates effective antimicrobial peptides that may be identified as potential antimicrobials [66,67,68]. We confirmed the inhibitory effect of CP on *S. aureus* to validate its antimicrobial activity. CP demonstrated the ability to inhibit the growth of *S. aureus*, indicating its property of delaying or impeding bacterial growth. The antimicrobial effect is mainly attributed to positively charged amino acids such as His, Lys, and Arg. Lys and Arg, in particular, can exhibit antimicrobial activity against bacteria and other microorganisms [69]. The antimicrobial effect of CP is likely due to these amino acids.

Inflammation leading to thickened skin is a characteristic lesion of AD, resulting from pathological inflammatory stimuli and impaired epidermal barrier function. Therefore, for the treatment of AD, we focused on restoring immune responses and preventing skin lesions in an AD mouse model. Local administration of CP demonstrated anti-AD effects in DNCB-induced mice. The in vivo efficacy of CP was tested using a DNCB-induced AD mouse model. DNCB sensitizes the epidermis similar to AD skin lesions, causing severe erythema, inflammation, scaling, and removal of the stratum corneum, allowing us to model the pathological features of AD. After local administration of CP, the recovery of skin lesions was consistently noticeable and significant compared to DNCB-induced mice until the end of the experiment. AD is an inflammatory skin disorder characterized by epidermal distribution and infiltration of immune cells. The inhibition of epidermal acanthosis and infiltration of inflammatory cells are pathological characteristics of AD skin lesions. Both features were visualized through histological analysis in the DNCB-induced model. CP suppressed epidermal acanthosis and infiltration of inflammatory cells (Figure 4). Cytokines secreted by Th1 and Th2 cells promote the development of AD [70,71]. Th2 cells predominate in allergic inflammation in the acute phase of AD, leading to increased expression of IL-6 [72]. In our study, IL-6 levels significantly increased in the serum of mice with DNCB-induced AD, while CP significantly reduced IL-6 production (Figure 5).

These results indicate that CP, an antioxidative and antimicrobial peptide derived from *C. notata* by-products, exhibits effective therapeutic effects in an AD mouse model. The discovery of novel peptides from *C. notata* by-products suggests new possibilities for utilizing by-products in the seafood processing industry. Moreover, the potential use of CP as a treatment for dermatitis through its antioxidative and antimicrobial effects is implied.

## 4. Materials and Methods

### 4.1. Experimental Materials

The following reagents were purchased from Sigma-Aldrich Chemical Co. (St. Louis, MO, USA) and used as received: ABTS, potassium persulfate, acetonitrile, dimethyl sulfoxide (DMSO), penicillin G, acetone, DNCB, and Terfenadine. Phosphate-buffered saline (PBS) was obtained from Biosesang Co. (Yongin-si, Gyeonggi-do, Republic of Korea) and used in the experiments. The enzymes Alcalase, Flavourzyme, Neutrase, and Protamex were purchased from Novozyme Co. (Bagsvaerd, Denmark). The 3500 Da dialysis membrane was acquired from Spectrum Labs Co. (Gardena, CA, USA) and utilized in the study. Acetonitrile was procured from J.T. Baker Co. (Phillipsburg, NJ, USA). Reagents for microbial culture, including nutrient broth and nutrient agar media, were sourced from BD DIFCO Co. (Franklin Lakes, NJ, USA). The IL-6 ELISA kit used was obtained from R&D Systems Co. (Minneapolis, MN, USA).

### 4.2. Preparation of Enzymatic Hydrolysate from C. notata By-Products

In this study, waste heads of *C. notata* were collected from the Fish Market on Jeju Island in Korea and utilized to prepare enzymatic hydrolysates for experimental samples. The collected heads of *C. notata* were freeze-dried using a freeze dryer (Vision, Daejeon, Republic of Korea) and subsequently ground into a powder using a grinder. The obtained powder was then utilized as a substrate for enzymatic hydrolysis for further processing. A mixture was prepared by adding 50 mL of distilled water at pH 7.0 to 1 g of the powdered *C. notata* heads, along with 20 mg/mL of enzymatic solution and 40 mM sodium sulfite (Na_2_SO_3_). The enzymatic hydrolysis process was carried out at 50 °C in a shaking incubator for 8 h, followed by termination of the enzymatic reaction at 100 °C. The resulting enzymatic hydrolysate was filtered using Whatman No. 41 filter paper (Whatman Ltd., Maidstone, Kent, UK), and the filtrate was then freeze-dried using a freeze dryer (Vision, Daejeon, Republic of Korea) to obtain a powdered form after grinding. The powder was stored in a freezer until further use. The enzyme and conditions used in the experiment are shown in Table 2.

### 4.3. Measurement of ABTS Radical-Scavenging Activity in Enzymatic Hydrolysates

The ABTS radical-scavenging activity of the enzymatic hydrolysate from *C. notata* heads was measured using the method adapted from Re et al.’s ABTS+· cation decolorization assay [73]. The ABTS radical solution used in the experiment was prepared by mixing 7 mM ABTS and 2.45 mM potassium persulfate in equal amounts and allowing the reaction to take place at room temperature (RT) in the dark for 24 h, resulting in the generation of ABTS+·. The generated ABTS radical solution was diluted with PBS (pH 7.4) to achieve an absorbance value of 0.70 (±0.02) at 732 nm. Enzymatic hydrolysates, diluted with PBS at various concentrations, were mixed with 180 μL of the prepared ABTS radical solution, and after a reaction time of 1 min, the absorbance was measured at 732 nm to assess the radical-scavenging activity.

### 4.4. Peptide Isolation from Alcalase Hydrolysate of C. notata By-Product

To isolate peptides from the high-ABTS activity Alcalase enzymatic hydrolysate of *C. notata* by-products, a 3500 Da dialysis membrane was initially employed to separate the sample inside and outside the membrane for one day. After freeze-drying, the sample was further fractionated at a flow rate of 0.5 mL/min using FPLC equipped with a desalting column to examine the molecular weight distribution. The sample outside the 3500 Da dialysis membrane, showing superior ABTS activity, was prepared at a concentration of 5 mg/mL. Using FPLC equipped with a GPC (SuperdexTM 30 Increase 10/300 GL) column at a flow rate of 0.5 mL/min, three fractions were obtained. These fractions were freeze-dried, and their ABTS radical-scavenging activity was measured. The most active CHAO-1 fraction was further separated into CHAO-1-I and CHAO-1-II fractions using HPLC with a C_18_ column (ZORBAX SB-C_18_, 4.6250 mm, 5 μm). After freeze-drying, the ABTS radical-scavenging activity was measured, revealing that CHAO-1-I exhibited superior activity. The highly active CHAO-1-I fraction was confirmed to be a single peak using HPLC with a GPC (YMC-Pack Diol-60, 4.6 × 250 mm, 5 μm) column.

### 4.5. Amino Acid Sequence Analysis and Synthesis

The amino acid sequence of the isolated peptide was determined using the automated Edman degradation method with the Protein Sequencer (PPSQ-51A, Shimadzu, Kyoto, Japan), which allowed for sequence analysis starting from the N-terminus.

### 4.6. S. aureus Cultivation

For this study, the bacterial strain used was *S. aureus* (ATCC 6538), obtained from the Korean Collection for Type Cultures. The growth media for bacterial cultivation consisted of nutrient broth and agar, which were sterilized at 121 °C for 15 min prior to use. The nutrient broth was prepared by dissolving 10 g of peptone, 5 g of beef extract, and 5 g of sodium chloride in 1000 mL of DW. The pH was adjusted to 7.2 ± 0.2. The agar medium was prepared by adding 15 g of agar to the above nutrient broth composition per liter. The entire mixture was then autoclaved for sterilization. Bacterial strains were cultured at 37 °C in an incubator for experimental purposes.

### 4.7. Antimicrobial Activity Measurement of Peptides Using the Agar Paper Disc Method

*S. aureus* cultured in nutrient broth medium was diluted onto agar plates to achieve an optical density at 600 nm (OD_600_) of 0.1 (corresponding to 1 × 10^8^ CFU/mL). Petri dishes (87 × 15 mm) were sterilized and filled with agar medium for the cultivation of *S. aureus*. The agar medium composition was prepared according to the detailed composition and concentrations referenced in Section 4.6. Subsequently, 100 μL of the diluted bacterial culture was evenly spread onto each plate. Sterilized paper discs (8 mm, Advantec) were then placed uniformly on the agar surface. Peptide solutions, prepared at different concentrations, were applied onto the paper discs in 20 μL volumes. The plates were then incubated at 37 °C for 24 h, and the antimicrobial activity was measured by assessing the size of the inhibition zone (mm) around the paper disc.

### 4.8. Determination of Minimum Inhibitory Concentration (MIC) of Peptides

To determine the MIC of the peptides, an adapted method based on Soares et al. [74] was employed. Solutions of the peptides at various concentrations were prepared in nutrient broth medium, and 180 μL of each solution was dispensed into wells 2 to 11 of a 96-well plate. Liquid growth medium (200 μL) was added to well 1, and 180 μL was added to well 12. Subsequently, *S. aureus* cultured in nutrient broth medium was diluted to achieve an optical density at 600 nm (OD_600_) of 0.1. Prior to measurement, the 96-well plate was shaken, and 20 μL of the diluted bacterial culture was added to wells 2 through 12 of the plate. The plate was then incubated at 37 °C, and absorbance readings at 490 nm were taken at 0, 2, 4, 6, 8, 12, and 24 h to monitor bacterial growth. The concentration at which bacterial growth was no longer observed was determined as the MIC value.

### 4.9. DNCB-Induced Atopic Dermatitis (AD) Mouse Model

In vivo experiments were conducted with the approval of the Institutional Animal Care and Use Committee (Approval: DKU-23-042) at Dankook University, following the guidelines for the care and use of experimental animals. Six-week-old male BALB/c mice were obtained from DBL (Eumseong, Republic of Korea) for the study. The mice were housed in a controlled animal facility with a temperature of 23 ± 2 °C and humidity of 50 ± 10% during the experiment. They were provided with standard chow and water for 10 days to stabilize before being divided into four groups. The groups included a normal group, an atopic dermatitis-induced group (DNCB), and two treatment groups post-atopic dermatitis induction, one treated with CP peptide (1 mg/200 μL) and the other with Terfenadine (0.1 mg/200 μL). Each group consisted of five mice.

Atopic dermatitis induction was performed using methods adapted from Kim et al. [75], Lee and Woo [76], and Jeon et al. [77]. A 3:1 mixture of acetone and olive oil containing 0.5% or 1% DNCB was prepared as the sensitizer solution. The samples were prepared using a solvent mixture of olive oil and PBS in a 1:9 ratio. After shaving and a 24 h stabilization period, the AD-induced group and the sample treatment groups were treated with 1% DNCB (200 μL) on the back once daily for three days. Subsequently, after a latent period of three days, a secondary sensitization was performed by applying 0.5% DNCB (200 μL) to the back three times a week for three weeks. The samples were applied concurrently with the secondary sensitization, 2 h after the application of 0.5% DNCB. The non-induced atopic dermatitis group was treated with a solvent mixture of olive oil and PBS (1:9). All mice were sacrificed on the 29th day of the experiment (Figure 6).

### 4.10. Histological Evaluation in the AD Animal Model

On the day of mouse sacrifice, skin tissue samples were collected from the four groups, fixed in 10% formalin solution, embedded in paraffin, and sliced into 4 μm thick sections. The sections were then subjected to Masson’s trichrome and toluidine blue staining to observe changes in the thickness of the epidermis and dermis, as well as infiltration of inflammatory cells [78].

### 4.11. Changes in Serum IL-6 Levels in the Atopic Dermatitis Animal Model

Blood collected on the day of mouse sacrifice was centrifuged at 13,000× *g*, 4 °C, for 15 min to separate the serum. The concentration of IL-6 in the serum was measured using a mouse ELISA kit. The experiment was conducted following the recommended procedures provided by the kit.

### 4.12. Statistical Analysis

The data are presented as mean ± standard deviation (SD). Statistical significance was analyzed using analysis of variance (ANOVA). *p* values < 0.05 were considered statistically significant.

## 5. Conclusions

This study investigated the antioxidant, antimicrobial, and anti-AD effects of a novel peptide, *C. notata* by-product peptide (CP), derived from the hydrolysate of *C. notata* by-product protein. The Alcalase hydrolysate demonstrated the highest antioxidant activity, and the subsequent purification process led to the isolation of CP, characterized by the amino acid sequence Ala-Gln-Val-Met-Lys-Leu-Pro-His-Arg-Met-Gln-His-Ser-Gln-Ser. The antimicrobial activity of CP against *Staphylococcus aureus* confirmed its potential as an effective antimicrobial peptide. The study also explored the therapeutic effects of CP on a mouse model of AD induced by DNCB. Local administration of CP significantly alleviated inflammatory skin lesions, reduced epidermal and dermal thickness, and inhibited mast cell infiltration in the dermis. Moreover, CP demonstrated an inhibitory effect on the elevated levels of IL-6 in the serum of DNCB-induced mice. These findings suggest that CP, derived from a *C. notata* by-product, possesses promising properties for potential applications in skincare and dermatological treatments. The discovery of novel peptides from by-products of seafood underscores potential for their eco-friendly utilization in the seafood processing industry. Further research and development could explore the utilization of CP as a therapeutic agent for skin conditions, leveraging its antioxidant, antimicrobial, and anti-inflammatory properties.

## Figures and Tables

**Figure 1 marinedrugs-22-00044-f001:**
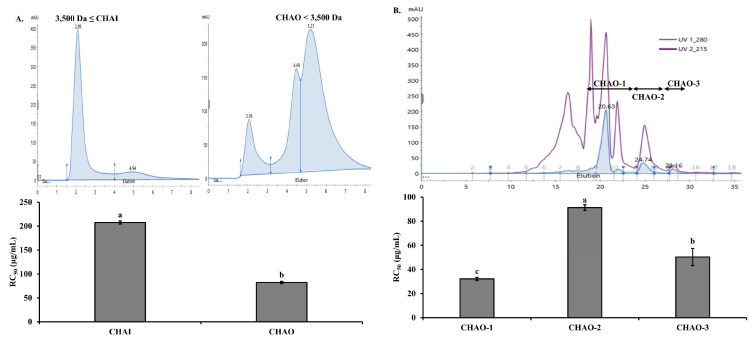
Purification of Alcalase hydrolysate. (**A**) FPLC pattern of hydrolysate by 3500 Da dialysis membrane, and the ABTS radical-scavenging activities (lower panel) of the fractions. (**B**) FPLC pattern on GPC column of the CHAO active fraction, and the ABTS radical-scavenging activities (lower panel) of the fractions. (**C**) Reversed-phase HPLC pattern of the CHAO-1 active fraction, and the ABTS radical-scavenging activities (lower panel) of the fractions. (**D**) HPLC pattern with a GPC column of the CHAO-1-I active fraction. All values are expressed as mean ± SD. Different letters are significantly different at *p* < 0.05 according to Duncan’s multiple range test.

**Figure 2 marinedrugs-22-00044-f002:**
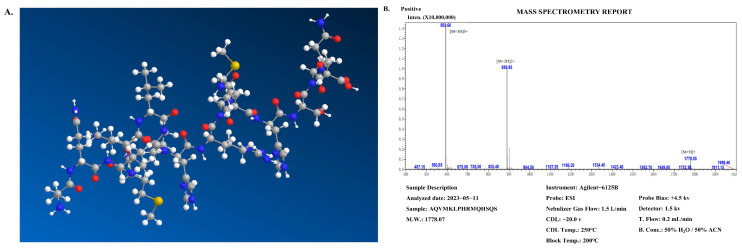
Structural characterization of *C. notata* by-product derived from the CP peptide. (**A**) Chemical structure of the CP peptide. (**B**) Synthesis results of *C. notata* by-product-derived CP peptide. (**C**) An example of a helical wheel diagram illustrated for the CP peptide, showing the net projection (left panel) and the wheel projection (right panel) for the CP peptide. (**D**) Antiparallel alignment of two helical wheels of the CP peptide.

**Figure 3 marinedrugs-22-00044-f003:**
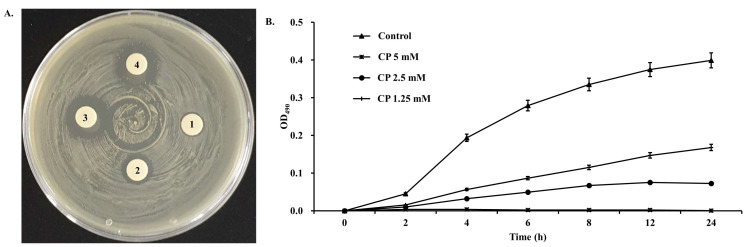
Antimicrobial activity of novel peptide CP. (**A**) Agar disk diffusion test; 1 = 1.25 mM CP; 2 = 2.5 mM CP; 3 = 5 mM CP; 4 = 25 ng penicillin. (**B**) Effect of CP on growth of *S. aureus.* OD_490_ = optical density at a wavelength of 490 nm.

**Figure 4 marinedrugs-22-00044-f004:**
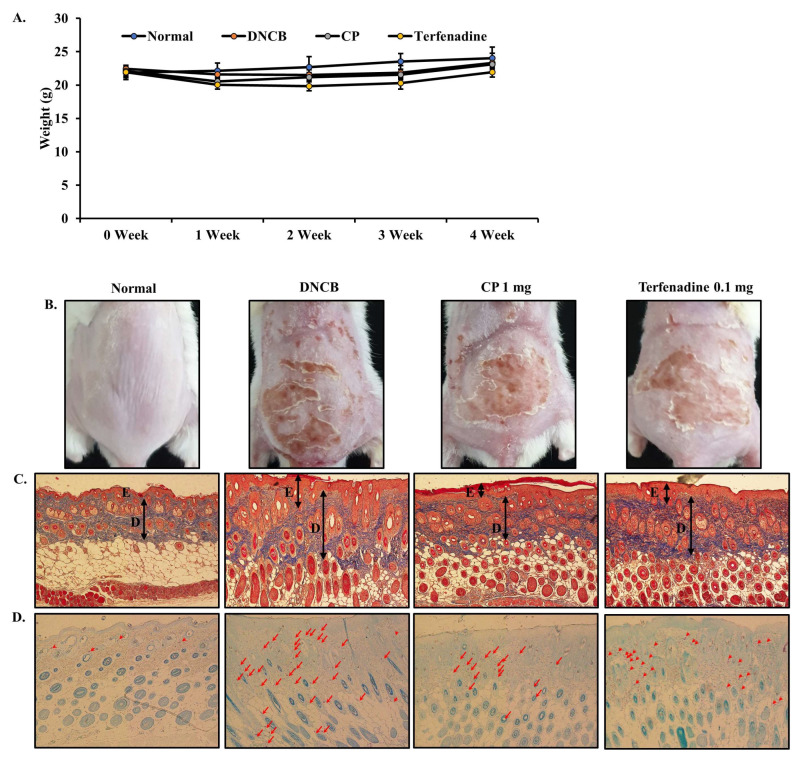
Effect of CP on body weight and histopathological changes in DNCB-induced AD mouse model. (**A**) Changes in body weight of DNCB-induced AD mice during the experiment. (**B**) Clinical severity of inflammatory skin lesions. Photographs were taken on day 28. (**C**,**E**,**F**) The thicknesses of the epidermis and dermis were examined by Masson’s trichrome staining of the skin sections. (**D**) The infiltration of mast cells in the dermis was examined by toluidine blue staining of the skin section. (**G**) The mast cells were counted in 3 fields. All results are shown as the mean ± SD (n = 5 per group). ## *p* < 0.01 compared to the control group. * *p* < 0.05 and ** *p* < 0.01, compared to the DNCB group. Normal, untreated group; DNCB, DNCB-sensitized group; CP, 1 mg CP-treated group; Terfenadine, 0.1 mg Terfenadine-treated group; E, epidermal thickness; D, dermal thickness.

**Figure 5 marinedrugs-22-00044-f005:**
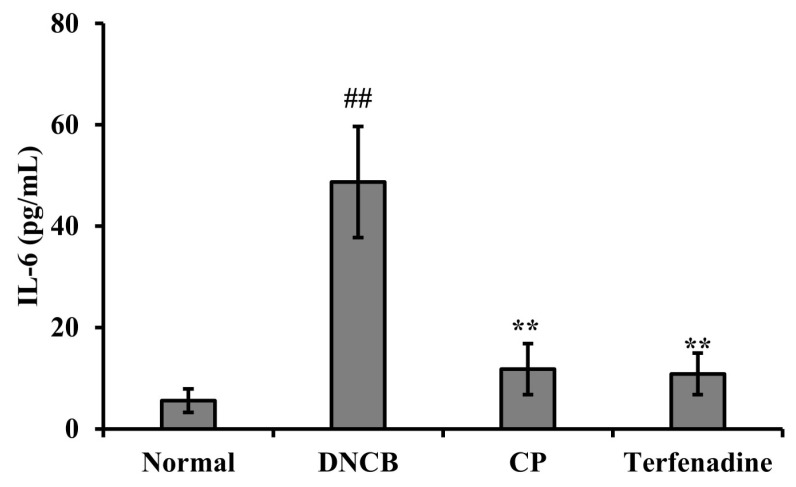
Inhibitory effect of CP on secreted IL-6 cytokines in the DNCB-induced mouse serum. All results are shown as the mean ± SD (n = 5 per group). ## *p* < 0.01 compared to the control group. ** *p* < 0.01 compared to the DNCB group. Normal, untreated group; DNCB, DNCB-sensitized group; CP, 1 mg CP-treated group; Terfenadine, 0.1 mg Terfenadine-treated group.

**Figure 6 marinedrugs-22-00044-f006:**
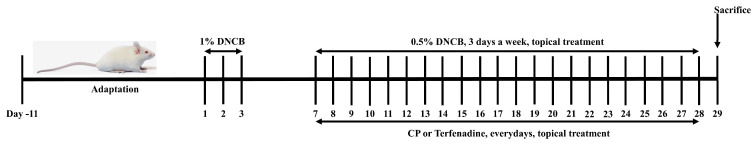
Experimental procedure of the model of DNCB-induced atopic dermatitis. The mice in the Normal group were treated with 9:1 PBS/olive oil. AD was induced by DNCB in BALB/c mice. Following initial sensitization with 1% DNCB, the BALB/c mice were dorsally treated with 0.5% DNCB 3 times a week for 3 weeks. CP and terfenadine preparations of 1 and 0.1 mg in 200 µL of PBS/olive oil (9:1) were applied on the dorsal skin of the mice in the CP and terfenadine groups every day for 21 days. Normal, untreated group; DNCB, DNCB-sensitized group; CP, CP 1 mg treated group; Terfenadine, Terfenadine 0.1 mg treated group.

**Table 1 marinedrugs-22-00044-t001:** Yields and RC_50_ values for ABTS radical-scavenging activities of four types of enzymatic extracts from *C. notata* by-product.

Enzymes	Alcalase	Flavourzyme	Neutrase	Protamex	Vit.C
Yield (%)	68.72	34.14	49.71	61.31	-
RC_50_ (μg/mL)	102.62 ± 5.63 ^d^	260.48 ± 3.40 ^b^	605.61 ± 220.58 ^a^	175.64 ± 6.98 ^c^	7.17 ± 2.02 ^e^

Vit. C was used as positive control. All values are expressed as mean ± SD. Different letters are significantly different at *p* < 0.05 according to Duncan’s multiple range test.

**Table 2 marinedrugs-22-00044-t002:** The optimum hydrolysis conditions of various enzymes.

Enzymes	pH	Temperature (°C)	Substrate:Enzyme
Alcalase	7.0	50.0	50:1
Flavourzyme	7.0	50.0	50:1
Neutrase	7.0	50.0	50:1
Protamex	7.0	50.0	50:1

## Data Availability

All data generated or analyzed during this study were included in this article.

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
