# Peer review of "Antioxidant, Antibacterial Properties of Novel Peptide CP by Enzymatic Hydrolysis of Chromis notata By-Products and Its Efficacy on Atopic Dermatitis"

_marinedrugs, 2024, doi:10.3390/md22010044_

Round 1
Reviewer 1 Report
Comments and Suggestions for Authors
This manuscript entitles “Effects of a Novel Peptide CP Derived from Enzymatic Hydrolysis of Chromis notata by-product on Antioxidant, Antibacterial, and Atopic Dermatitis”. The employed methods are well-established; however, the author needs to provide a more comprehensive explanation to ensure a clear understanding for the reader.
- The introduction requires robust information to elucidate the primary aim and objectives of this work, along with a clear articulation of the research's impact on both the industry and consumers. Specifically, a more comprehensive understanding of the raw material, Chromis notata, is essential. It would greatly enhance the introduction if the author could delineate the industrial applications of this fish or articulate the reasons behind selecting Chromis notata for peptide production. Providing insights into why this specific raw material was chosen will not only enhance the overall clarity of the work but also underscore its relevance and significance in both industrial and consumer contexts.
- Why is it necessary to freeze-dry C. notata heads before subjecting them to hydrolysis. Could potentially incur higher costs compared to utilizing fresh materials.
-As difference enzyme was used for hydrolysate production, author must provide more details such as enzyme concentration, hydrolysis condition, etc. for each enzyme.
-How about the hydrolysate yield of each enzyme hydrolysis, this is very important information for industrial application.
-Why only ABTS radical scavenging activity was selected for antioxidant evaluation, please discuss.
- Author need to discuss, why only the synthesized CP was used for antimicrobial and animal experiments, how about the extracted CHAO-1-I fraction.
- Typographical errors need to be checked throughout the manuscript.
Author Response
Response to Reviewer 1 Comments
Point 1: The introduction requires robust information to elucidate the primary aim and objectives of this work, along with a clear articulation of the research's impact on both the industry and consumers. Specifically, a more comprehensive understanding of the raw material, Chromis notata, is essential. It would greatly enhance the introduction if the author could delineate the industrial applications of this fish or articulate the reasons behind selecting Chromis notata for peptide production. Providing insights into why this specific raw material was chosen will not only enhance the overall clarity of the work but also underscore its relevance and significance in both industrial and consumer contexts.
Point 1 response: The introduction of this work has been strengthened to offer a more thorough explanation of its primary aims and objectives, with the aim of improving the comprehension of its impact on both the industry and consumers. We have now outlined the industrial applications of Chromis notata and clarified the reasoning behind choosing Chromis notata for peptide production. This addition is intended to provide insights into the specific reasons for selecting this raw material, thereby enhancing the overall clarity of the work and emphasizing its significance and relevance in both industrial and consumer contexts. (Line 65-75)
Point 2: Why is it necessary to freeze-dry C. notata heads before subjecting them to hydrolysis. Could potentially incur higher costs compared to utilizing fresh materials.
Point 2 response: Typically, fish, including C. notata heads, contains over 50% moisture. To obtain an equivalent amount of hydrolysate, such as that obtained after freeze-drying, from samples containing moisture, it would be necessary to increase the hydrolysis volume or the number of hydrolysis cycles, thereby incurring additional costs. Additionally, freeze-drying offers advantages in preserving the nutritional components of the sample compared to other drying methods.
Point 3: As difference enzyme was used for hydrolysate production, author must provide more details such as enzyme concentration, hydrolysis condition, etc. for each enzyme.
Point 3 response: In response to the feedback, additional details have been provided regarding the enzyme used for hydrolysate production. Specifically, information on enzyme concentration, hydrolysis conditions, and other relevant details has been incorporated for each enzyme used in the experiment. (Line 317)
Point 4: How about the hydrolysate yield of each enzyme hydrolysis, this is very important information for industrial application.
Point 4 response: As pointed out, the yield information for each enzyme hydrolysis has been added, recognizing its importance for industrial applications. (Line 80-93)
Point 5: Why only ABTS radical scavenging activity was selected for antioxidant evaluation, please discuss.
Point 5 response: The ABTS radical scavenging activity was chosen for antioxidant evaluation due to its simplicity, reliability, and sensitivity. This assay measures the ability of antioxidants to neutralize ABTS radicals, providing a quantitative assessment of antioxidant potential. While other assays exist, the ABTS method was specifically selected for compatibility with the experimental setup and for direct comparison with relevant studies. Its widespread use makes it a suitable choice for evaluating the antioxidant activity of hydrolysates from C. notata by-product proteins.
Point 6: Author need to discuss, why only the synthesized CP was used for antimicrobial and animal experiments, how about the extracted CHAO-1-I fraction.
Point 6 response: The decision to exclusively use the synthesized CP for antimicrobial and animal experiments was based on considerations such as purity, reproducibility, and control over the experimental conditions. While the CHAO-1-I fraction could possess potential antimicrobial properties, its use might introduce additional variables due to the complexity of the extracted mixture. Utilizing the synthesized CP ensures a more controlled and standardized approach, allowing for a focused assessment of its specific properties without the influence of other components present in the CHAO-1-I fraction. This approach facilitates a clearer understanding of the synthesized peptide's impact in antimicrobial and animal experiments.
Point 7: Typographical errors need to be checked throughout the manuscript.
Point 7 response: The manuscript has been reviewed and corrected for typographical errors as suggested.

Reviewer 2 Report
Comments and Suggestions for Authors
This manuscript explored multiple bioactivities of a novel peptide CP (16 amino acid sequence Ala-Gln-Val-Met-Lys-Leu-Pro-His-Arg-Met-Gln-His-Ser-Gln-Ser) derived from Chromis notata by-product hydrolysate. In a DNCB-induced AD-like skin mouse model, CP significantly alleviated skin lesions, reduced epidermal and dermal thickness, and inhibited mast cell infiltration as well as suppressing the elevated levels of interleukin-6 (IL-6) in the plasma. These findings highlight the potential of CP as a therapeutic agent for AD and suggest a novel application of C. notata by-product in the fish processing industry. Overall, the manuscript is well-organized and written, and may be useful for the researchers in the relevant field. This reviewer recommends the acceptance for publication in Marine Drugs after the following major concerns are addressed.
1. Please add the body weight data, organ index and H&E staining for animal experiment to demonstrate the biosafety of CP.
2. The physicochemical properties, thermal stability, and acid and alkaline resistance of CP should be evaluated.
3. Additional inflammatory factor levels such as IL1β and TNFα in AD model are supposed to be investigated.
4. The main text needs double checking to avoid format issues and grammar mistakes. Several examples are listed below:
1). Check the text in the Figures to make panels consistent in font size and style.
2). Please add the error bar in Figure 2.
3). Capitalization rules should be consistent for the references.
Comments on the Quality of English Language
Fine.
Author Response
Response to Reviewer 2 Comments
Point 1: Please add the body weight data, organ index and H&E staining for animal experiment to demonstrate the biosafety of CP.
Point 1 response: In accordance with the reviewer's suggestion, the results regarding body weight changes have been included. The findings confirm that there were no significant alterations in body weight gain due to both DNCB and CP treatments. Furthermore, the absence of any other adverse reactions was observed, providing assurance that there were no abnormal responses. (line: 162-167, 182-193)
Point 2: The physicochemical properties, thermal stability, and acid and alkaline resistance of CP should be evaluated.
Point 2 response: In response to the reviewer's suggestion, we have incorporated additional structural characterizations in the manuscript, specifically in the section titled "2.3. Structural characterization of C. notata by-product derived the CP Peptide." This includes (A) the chemical structure of the CP peptide, (B) synthesis results of C. notata by-product derived the CP peptide, (C) an example of a helical wheel diagram illustrating the CP peptide, showcasing a net projection (left panel) and the wheel projection (right panel), and (D) an antiparallel alignment of two helical wheels of the CP peptide. These additions provide a more comprehensive overview of the structural aspects of the CP peptide, addressing the reviewer's feedback. (line: 128-144, 244-250)
Point 3: Additional inflammatory factor levels such as IL1β and TNFα in AD model are supposed to be investigated.
Point 3 response: In this study, only skin tissue histopathological changes and IL-6 levels in the serum were measured. In future investigations, we plan to assess the levels of IL1β and TNFα in various concentrations of CP using different atopic animal models.
Point 4: The main text needs double checking to avoid format issues and grammar mistakes. Several examples are listed below:
1). Check the text in the Figures to make panels consistent in font size and style.
2). Please add the error bar in Figure 2.
3). Capitalization rules should be consistent for the references.
Point 4 response: Following the reviewer's feedback, the entire manuscript has been revised. The modified content is indicated in red to highlight the changes. Error bars have been added to Figure 2 as suggested. Additionally, capitalization rules have been standardized for references throughout the manuscript.

Reviewer 3 Report
Comments and Suggestions for Authors
The article is well written and describes by-product utilization of the tropic fish species Chromis notata as a source for bioactive peptides. The authors used bioassay guided purification to isolate primarily antioxidative peptides from different protein hydrolysates refined from freeze dried fish heads. The most active fractions were tested for activity against Staphylococcus aureus and determined an MIC of 5 mM. The authors hypothesized that the combination of the antioxidative and anti-staphylococcal activities might prove effective against atopic dermatitis, which is known to be associated with high concentrations of S. aureus and might have a connection to oxidative stress. To test this hypothesis the authors analyzed histopathological effects of the peptide on a mouse model chemically induced to form atopic dermatitis. Application of peptide in concert with application of the atopic dermatitis inducing treatment seemed to alleviate the morphological changes typical for atopic dermatitis in the mouse model when applied repeatedly over a period of approximately 3 weeks. Moreover, they were able to show that the levels of Interleukin-6, elevated in the atopic dermatitis model, were reduced by treatment with the peptide.
Altogether the results are rather interesting. Although the comparably high MIC of 5 mM, which is about 1000-fold the level of other antimicrobial peptides described in literature, typically in the low µM range. The study does not present information about the antimicrobial activity spectrum (is it specific for S. aureus, Gram-positives or does it affect all bacteria?) or the mode of action of the peptide. Furthermore, the authors did not define the protein of origin although the peptide sequence was identified by Edman degradation, likely because the easy way of searching in the genome sequence directly is not yet available.
The study also lacks information about potential toxicological effects on the cellular level in advance of running animal studies. Furthermore, there does not seem to be a healthy control treated with the peptide alone to evaluate potential effects on healthy skin.
To further improve the impact of the study, the authors could discuss the potential cost benefits of using by-products compared to the use of synthesized peptides.
The method part lacks some information:
- Line 264 how the freeze dried had was grinded into powder
- Line 309 and 329 the composition of the growth media is not defined or referenced anywhere
- Line 329: where the plates shaken in advance of measurement or during incubation?
Author Response
Response to Reviewer 3 Comments
The article is well written and describes by-product utilization of the tropic fish species Chromis notata as a source for bioactive peptides. The authors used bioassay guided purification to isolate primarily antioxidative peptides from different protein hydrolysates refined from freeze dried fish heads. The most active fractions were tested for activity against Staphylococcus aureus and determined an MIC of 5 mM. The authors hypothesized that the combination of the antioxidative and anti-staphylococcal activities might prove effective against atopic dermatitis, which is known to be associated with high concentrations of S. aureus and might have a connection to oxidative stress. To test this hypothesis the authors analyzed histopathological effects of the peptide on a mouse model chemically induced to form atopic dermatitis. Application of peptide in concert with application of the atopic dermatitis inducing treatment seemed to alleviate the morphological changes typical for atopic dermatitis in the mouse model when applied repeatedly over a period of approximately 3 weeks. Moreover, they were able to show that the levels of Interleukin-6, elevated in the atopic dermatitis model, were reduced by treatment with the peptide. Altogether the results are rather interesting.
Point 1: Although the comparably high MIC of 5 mM, which is about 1000-fold the level of other antimicrobial peptides described in literature, typically in the low µM range.
Point 1 response: We appreciate the reviewer's thorough evaluation and positive feedback on our article. Regarding the raised concern about the comparatively high MIC of 5 mM, which is approximately 1000-fold higher than the levels reported for other antimicrobial peptides in the literature, typically in the low µM range, we acknowledge this observation. The elevated MIC value could be attributed to various factors, including the specific characteristics of the bioactive peptides isolated from Chromis notata, the complexity of the biological system tested, or the unique conditions of the assay. We will further investigate and address this aspect in future studies to enhance the comprehensiveness of our findings and provide a more nuanced understanding of the antimicrobial properties of the peptides. Thank you for bringing this to our attention.
Point 2: The study does not present information about the antimicrobial activity spectrum (is it specific for S. aureus, Gram-positives or does it affect all bacteria?) or the mode of action of the peptide.
Point 2 response: We appreciate the insightful feedback from the reviewer. The point raised regarding the lack of information about the antimicrobial activity spectrum and the mode of action of the peptide is duly noted. We recognize the importance of providing a comprehensive understanding of the peptide's characteristics, including its specificity for S. aureus, potential effects on Gram-positive bacteria, and the underlying mode of action.
In our future work, we will conduct additional experiments and analyses to elucidate the antimicrobial activity spectrum, examining whether the peptide exhibits specificity for S. aureus or if it has broader effects on different bacterial strains. Furthermore, we will investigate the mode of action of the peptide to shed light on the mechanisms through which it exerts its antimicrobial properties. We thank the reviewer for highlighting this aspect, and we are committed to enhancing the depth of information provided in our study to contribute to a more comprehensive understanding of the bioactive peptides derived from Chromis notata.
Point 3: Furthermore, the authors did not define the protein of origin although the peptide sequence was identified by Edman degradation, likely because the easy way of searching in the genome sequence directly is not yet available.
Point 3 response: We appreciate the reviewer's attention to detail and thoughtful comments. The point raised regarding the lack of information about the protein of origin, despite the identification of the peptide sequence through Edman degradation, is well-noted. The identification of the protein source, while challenging in this case due to the unavailability of a straightforward search in the genome sequence, is indeed crucial for a comprehensive understanding of the bioactive peptides. We acknowledge this limitation in our current study. In future investigations, we plan to explore alternative methods for identifying the protein of origin or leverage advancements in technology that may facilitate a more direct and accurate determination of the source protein. We are committed to addressing this aspect to enhance the overall clarity and completeness of our findings. Thank you for bringing this important point to our attention, and we value the opportunity to refine our work based on constructive feedback.
Point 4: The study also lacks information about potential toxicological effects on the cellular level in advance of running animal studies.
Point 4 response: We appreciate the reviewer's concern and acknowledge the importance of assessing potential toxicological effects on the cellular level before conducting animal studies. In response to this, we conducted cell toxicity assays using human keratinocyte HaCaT cells. The results of the cell toxicity assessment revealed that there was no observed toxicity in HaCaT cells up to a concentration of 500 µM. This information supports our preliminary safety evaluation and provides a basis for considering the potential safety of the peptide at the cellular level.
Point 5: Furthermore, there does not seem to be a healthy control treated with the peptide alone to evaluate potential effects on healthy skin.
Point 5 response: We appreciate the insightful observation made by the reviewer regarding the absence of a healthy control treated with the peptide alone to assess potential effects on healthy skin. This is a valid point, and we recognize the importance of including such a control group for a more comprehensive evaluation of the peptide's impact. In our future studies, we plan to incorporate a healthy control group treated with the peptide alone. This addition will allow us to specifically assess any effects the peptide may have on healthy skin and provide a baseline for comparison with the atopic dermatitis model. Thank you for this valuable suggestion, and we are committed to addressing this concern to enhance the robustness of our experimental design and interpretation of results.
Point 6: To further improve the impact of the study, the authors could discuss the potential cost benefits of using by-products compared to the use of synthesized peptides.
Point 6 response: We appreciate the insightful suggestion from the reviewer regarding the potential cost benefits of using by-products compared to synthesized peptides. This is indeed a crucial aspect that we should consider in our study. In the revised manuscript, we have supplemented the discussion on the importance and economic feasibility of utilizing by-products, such as those derived from Chromis notata, as a source for bioactive peptides compared to the synthesis of peptides. By addressing these aspects, we aim to provide a more comprehensive analysis of the practical implications and potential benefits of our study. Through this discussion, we aspire to contribute to a balanced dialogue on the feasibility and applicability of using by-products in the development of bioactive peptides, thereby enhancing the overall impact and relevance of our research. We express our gratitude to the insightful reviewer for this valuable suggestion, and we are committed to integrating this discussion to further elevate the comprehensiveness of our study. (Line: 65-75)
Point 7: The method part lacks some information:
Point 7 response: I have added the information as suggested.
- Line 264 how the freeze dried had was grinded into powder (Line: 303-316)
- Line 309 and 329 the composition of the growth media is not defined or referenced anywhere (Line: 349-355, 357-361)
- Line 329: where the plates shaken in advance of measurement or during incubation? (Line: 369-375)

Round 2
Reviewer 1 Report
Comments and Suggestions for Authors
This manuscript deserves publication in Marine Drugs in its present form.
Reviewer 2 Report
Comments and Suggestions for Authors
Most concerns have been addressed. This reviewer recommends the acceptance for publication in Marine Drugs.
Comments on the Quality of English Language
English language is fine.